# Microbiomes of three coral species in the Mexican Caribbean and their shifts associated with the Stony Coral Tissue Loss Disease

Zita P. Arriaga-Piñón[1], J. Eduardo Aguayo-Leyva[2], Lorenzo Álvarez-Filip[3], Anastazia T. Banaszak[3], Ma. Leopoldina Aguirre-Macedo[1], David A. Paz-García[2]*, José Q. García-Maldonado[1]*

**1** Departamento de Recursos del Mar, Centro de Investigación y Estudios Avanzados del Instituto Politécnico Nacional, Unidad Mérida, Mérida, Yucatán, México, **2** Laboratorio de Genética para la Conservación. Centro de Investigaciones Biológicas del Noroeste (CIBNOR), La Paz, B.C.S., México, **3** Unidad Académica de Sistemas Arrecifales, Instituto de Ciencias del Mar y Limnología, Universidad Nacional Autónoma de México, Puerto Morelos, Quintana Roo, México

* jose.garcia@cinvestav.mx (JQGM); dpaz@cibnor.mx (DAPG)

## Abstract

Stony Coral Tissue Loss Disease (SCTLD) has caused widespread coral mortality in the Caribbean Region. However, how the disease presence alters the microbiome community, their structure, composition, and metabolic functionality is still poorly understood. In this study, we characterized the microbial communities of the tissues of apparently healthy and diseased SCTLD colonies of the species *Siderastrea siderea*, *Orbicella faveolata*, and *Montastraea cavernosa* to explore putative changes related to the presence of SCTLD. *Gammaproteobacteria*, *Alphaproteobacteria*, and *Bacteroidia* were the best represented classes in the healthy tissues of all coral species, and alpha diversity did not show significant differences among the species. The microbial community structure between coral species was significantly different (PERMANOVA: F = 3.46, p = 0.001), and enriched genera were detected for each species: *Vibrio* and *Photobacterium* in *S. siderea*, *Spirochaeta2* and *Marivivens* in *O. faveolata* and SAR202_clade and *Nitrospira* in *M. cavernosa*. Evidence of SCTLD in the microbial communities was more substantial in *S. siderea*, where differences in alpha diversity, beta diversity, and functional profiles were observed. In *O. faveolata*, differences were detected only in the community structure, while *M. cavernosa* samples showed no significant difference. Several microbial groups were found to have enriched abundances in tissue from SCTLD lesions from *S. siderea* and *O. faveolata*, but no dominant bacterial group was detected. Our results contribute to understanding microbial diversity associated with three scleractinian coral species and the shifts in their microbiomes associated with SCTLD in the Mexican Caribbean.

## Introduction

The coral microbiome is a complex and dynamic community of microorganisms that plays different but crucial roles, such as promoting host defense and health, nutrient acquisition,

**Data Availability Statement:** The raw sequencing data generated in this study have been deposited at

NCBI under the PRJNA1112859 BioProject accession number.

**Funding:** This research was supported by Consejo Nacional de Ciencia y Tecnología (CONACYT) through grant FORDECYT-PRONACES, CF-2019-425888 to A.T.B., D.P.-G. and J.Q.G.-M. We thank CONAHCYT for providing the doctoral scholarship to ZPAP (830311) and JEAL (1081013) during the development of this study. The funding institution, Consejo Nacional de Ciencia y Tecnología (CONACYT), had no role in study design, data collection and analysis, decision to publish, or preparation of the manuscript.

**Competing interests:** The authors have declared that no competing interests exist.

and environmental acclimation [1]. Several studies have characterized the structure and composition of bacterial communities associated with different coral species, life stages, and health states. These studies have shown that the microbiome is diverse and species-specific [2, 3], but also flexible and adaptable to geography, season, the surrounding environment, and host physiology and health [1]. Understanding the microbiome's composition, structure, and functional roles and how it has a wide impact on the survival of coral reefs facing anthropogenic stressful conditions, climate change, and coral disease outbreaks is an ongoing research field.

Coral diseases have been reported since the 1970s and have increased in number, affecting a wide range of coral species across all geographic regions [4]. In September 2014, stony coral tissue loss disease (SCTLD) was reported off the coast of Miami, Florida [5], and resulted in massive coral mortality due to the spread of this unprecedented disease-related event throughout the Caribbean. SCTLD is now considered one of the most lethal disease outbreaks ever recorded on coral reefs due to its high prevalence, the number of susceptible species, and the high mortality of affected corals [6]. In Mexico, SCTLD was first reported in July 2018 in Puerto Morelos, Quintana Roo, in the Mexican Caribbean [7]. Since then, the coral communities have been monitored to describe the impact of SCTLD on coral communities' composition and cover, disease prevalence, and coral-related mortality [7–10].

Although SCTLD affects more than 20 species [7], the effects on abundant species that contribute significantly to reef accretion are of particular importance. Among these are *Siderastrea siderea*, *Orbicella faveolata*, and *Montastraea cavernosa*, which are common species in the Mexican Caribbean, have similar life-history strategies [11], and seem to be stress-tolerant, as they persist in sites with nutrient and pollutant disturbance [12]. *O. faveolata* is a slow-growing species found on the crests and slopes of coral reefs and back reefs, while *M. cavernosa* and *S. siderea* are fast-growing species associated with the bottom of the reefs and sediment. Although all three species are considered intermediately susceptible species by the case definition of SCTLD from NOAA [13], SCTLD progresses faster in *S. siderea* than in *M. cavernosa* [14, 15].

SCTLD has been studied mainly in Florida to describe disease spread [15–17] and to develop the case definition [13], or in experimental approaches to elucidate virulence [4, 14] and the transmission process [14]. However, crucial information such as the causal pathogen, or its effects on the host microbiome, composition, structure, and metabolic functions, is still missing. It is believed that the pathogen could be of bacterial origin [4, 18] since antibiotics, such as amoxicillin or kanamycin, stop or slow down the progression of lesions. However, it has also been proposed that it could be viral [19], causing dysfunction and symbiophagy of the *Symbiodiniaceae* algae; therefore, algal identity directly influences the severity of the disease. Some studies [15, 16, 18, 20] have reported shifts in the structure of bacterial communities on the coral microbiome, and identified groups whose abundances increased during the onset of SCTLD. However, a notable gap exists in the analysis of functional potential, even through predictive methods.

In the Mexican Caribbean, there are just a few published studies of the coral microbiome [21, 22]. The knowledge of the composition and structure of the coral microbiome in different healthy coral species and monitoring how it changes in response to stress and disease, such as the SCTLD, is fundamental and critical for developing better strategies for protecting and restoring coral reefs. In this study, we first characterized the bacterial community structure, composition, and potential metabolic pathways in apparently healthy tissues from three reef-building coral species. We then explored the putative differences between microbial communities of healthy and SCTLD tissues in each coral species to evaluate their specific response in the diversity, composition, structure, and functional predictions to understand the changes of microbial communities because of the presence of this disease.

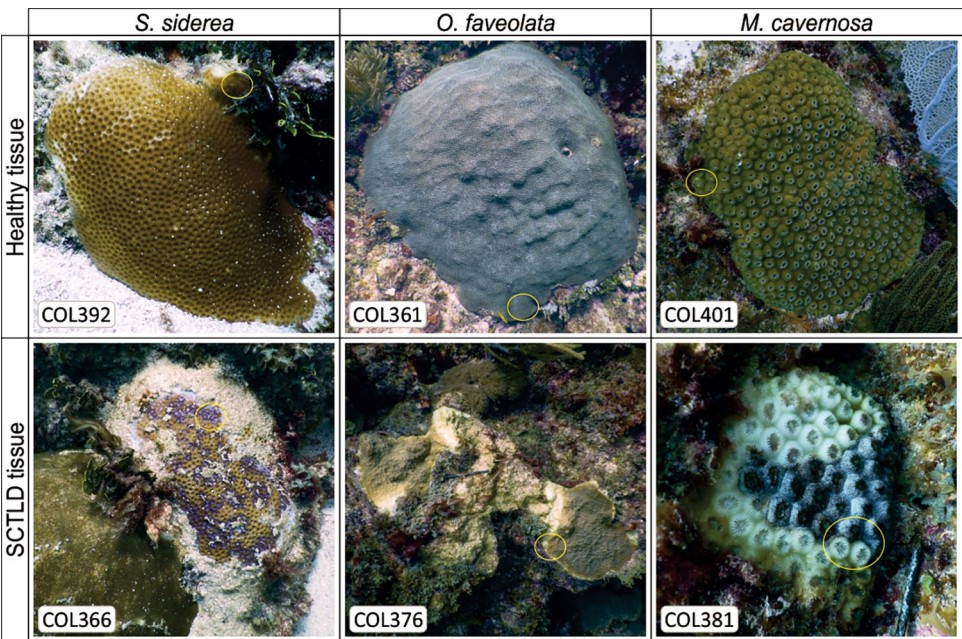

**Fig 1. Representative images showing colonies of *Siderastrea siderea*, *Orbicella faveolata*, and *Montastraea cavernosa*, with apparently healthy tissue (top row) and those affected by SCTLD lesion (bottom row).**

## Methods

### Sample collection

Samples of three coral species were taken from apparently healthy colonies (hereafter healthy tissue, n = 18) and colonies showing signs of the SCTLD lesion (hereafter SCTLD tissue, n = 22), from reefs in Puerto Morelos and Tulum, Quintana Roo, Mexico (Fig 1, S1_Table1 and S2_Fig1 in S1 File) during August 2020, under collection permit issued to Universidad Nacional Autónoma de México (PPF/DGOPA-070/20). The identification of diseased colonies was based on the appearance of lesions according to the case description of SCTLD [13]. Coral fragments were collected by SCUBA diving using a hammer and chisel or pruning shears. Fresh gloves were used to collect each sample. To prevent cross-contamination, healthy tissue samples were collected first, before SCTLD tissue samples. Samples were placed in individual bags, stored in RNAlater (Sigma R 0901, St Louis, MO), and transported to the lab.

### Microbiome sample processing and sequencing

DNA isolation from 0.2g of sample was performed using the DNeasy Blood & Tissue Kit Protocol (Qiagen, Hilden, Germany) according to the manufacturer's instructions. DNA quality was evaluated using agarose gel (1%) electrophoresis. Library preparation per sample for sequencing was made following the Illumina Protocol with a two-step PCR approach. Briefly, 16S rRNA V3 and V4 regions were amplified by PCR using the 16S V3 (341F) forward and V4 (805R) reverse primer pairs with added Illumina adapter overhang nucleotide sequences (S3_Table2 in S1 File) [23, 24] using 2μl of DNA (~20ng μl$^{-1}$), 0.5μl of each primer (10mM) and 10μl of 2x Phusion Flash High-Fidelity MasterMix (Thermo Scientific, Waltham, MA, USA). The PCR consisted of an initial denaturation at 95°C/3min, 28 cycles of 95°C/30s, 55°C/30s, 72°C/30s, and a final extension at 72°C/5 min. Amplicons were purified with AMPure XP Beads kit and labeled with the Nextera XT kit during a second PCR consisting of

8 cycles in the same thermo program. Indexed PCR products were purified again. Amplicon size was verified (~500bp in the 1st PCR and ~550bp in the 2nd PCR) in 1% agarose gel electrophoresis. Concentration was quantified with a Qubit 3.0 Fluorometer using the Qubit dsDNA HS Assay Kit (Life Technologies, Carlsbad, CA, USA). Sequencing libraries for the 16S rRNA gene were prepared according to the manufacturer's protocol. Paired-end sequencing (2x250 bp) was conducted using the MiSeq platform (Illumina, San Diego, CA, USA) with a MiSeq Reagent Kit V2 (500 cycles). The raw sequencing data generated in this study have been deposited at NCBI under the PRJNA1112859 BioProject accession number.

### Data quality control and statistical analysis

Paired-end reads (2x250) in the fastq format were processed with the QIIME2 (v. 2022.2) pipeline. The error correction and denoising to resolve the amplicon sequence variants (ASVs, a taxonomic unit for amplicon-based studies) [25] were performed with the DADA2 plugin (denoise-paired method) [26]. The representative ASVs were taxonomically assigned with the V-SEARCH consensus taxonomy classifier plugin [27] using the SILVA database (v.138) as reference. The information was loaded onto the R environment, and statistical analysis and visualization were performed with the phyloseq [28], vegan [29], and ggplot2 [30] libraries. The data set was normalized with the phyloseq library to obtain abundance tables, Shannon index, and richness values as observed ASVs.

Double principal coordinate analysis (DPCoA) was applied for the ordination of the microbial communities, with the relative abundance table and a distance matrix obtained from the square root of the cophenetic/patristic (cophenetic.phylo) distance between ASVs [28, 31]. Permutational multivariate analysis of variance (PERMANOVA) was used to test significant differences according to coral species and tissue condition (healthy or SCTLD tissue), using the "adonis" function with 999 permutations [32–34]. *Post-hoc* pairwise PERMANOVA tests were performed to identify significant differences between pairs of species. The linear discriminant analysis (LDA) effect size method (LEfSe) was applied to determine the clades most likely to explain differences between bacterial communities [35]. Finally, the functional potential of microbial communities was predicted using Phylogenetic Investigation of Communities by Reconstruction of Unobserved States 2 (PICRUSt2) [36]. Descriptions were determined by the pathway classification in the MetaCyc database. The differences in functional profiling were characterized via LEfSe analysis.

## Results

### Microbial community diversity and composition among healthy tissues of the coral species

In total, 6,148 ASVs were obtained from 18 healthy tissue samples across three reef-building species in the Central Caribbean of Mexico (*S. siderea* = 1,542, *O. faveolata* = 2,500 and *M. cavernosa* = 2,631 ASVs, S1_Table1 in S1 File). The microbial community diversity was similar among coral species healthy tissues (Shannon index: F = 1.25, p = 0.31 and richness: F = 1.25, p = 0.28, S1_Table1 in S1 File). The best represented classes in microbial community composition were *Gammaproteobacteria*, *Alphaproteobacteria*, and *Bacteroidia*. Classes with a high relative abundance were also *Planctomycetes*, *Acidimicrobiia*, and *Verrucomicrobia* (Fig 2A). The composition bar from each sample (healthy and SCTLD tissue) is shown in S4_Fig2 (Classes) and S5_Fig3 (Orders) in S1 File.

The microbial community structure significantly differed among healthy coral species (PERMANOVA: F = 3.4617, p = 0.001, S6_Fig4 in S1 File) and the *post-hoc* results indicated

significant differences among each species (S7_Table3 in S1 File). A total of 21 microbial clades had differential abundances across the three studied species (Fig 2b). *Vibrio*, *Photobacterium*, *Pseudoalteromonas*, *Algicola*, *Psychrosphaera*, *Acrophormium_PCC-7375*, and *Woesearchaeales* were the clades with a greater effect on community structure in *S. siderea*, whereas in *O. faveolata* the clades were *Spirochaeta_2*, Candidatus_*Amoebophilusand*, and *Marivivens*; and in *M. cavernosa* they were SAR202_clade, *Nitrospira*, and *Spirochaeta*. A total of 391 functional pathways were detected by functional prediction across all healthy tissues. Although no significant difference was found among predicted metabolisms from each coral species (PERMANOVA: F = 0.7862, p = 0.564), 45 pathways were found with differential abundance in each species (S8_Table4 in S1 File).

## Microbial community comparison between healthy and SCTLD tissue

When healthy and SCTLD tissues were compared across the three coral species, the relative abundance of some bacterial classes decreased in SCTLD tissues, such as *Dadabacteriia*, *Nanoarchaeia*, *Parcubacteria*, and Subgroup_22; while others exhibited higher relative abundances, such as *Actinobacteria*, *Bacilli*, *Campylobacteria*, *Clostridia*, *Desulfobacteria*, *Desulfovibrionia*, *Fusobacteriia*, and *Nitrospiria* (S4_Fig2 in S1 File).

## Siderastrea sidereal

The microbial diversity in *S. siderea* differed significantly between healthy and SCTLD tissues (Shannon index: F = 6.93 p = 0.03, Fig 3A; and richness: F = 5.81 p = 0.03, S1_Table1 in S1 File). The microbial structure also differed significantly (PERMANOVA: F = 22.92, p = 0.008, Fig 3B). The enriched genera in SCTLD tissue samples from *S. siderea* were *Vibrio*, *Photobacterium*, and *Synechococcus*_CC9902. In contrast, in healthy tissue samples, the most abundant were *Woeseia*, *Tistlia*, NB1−j k, and Pir4_lineages (Fig 4A). Significant changes were detected in the metabolic potential with 140 differential pathways between healthy tissues and SCTLD tissues (PERMANOVA: F = 6.436, p = 0.011). The most enriched pathways in healthy tissues were related to aromatic compound degradation, c1 compound utilization and assimilation, glycolysis, inorganic nutrient metabolism, and respiration. In contrast, the pathways enriched in SCTLD tissues were related to amine and polyamine biosynthesis, metabolic regulator biosynthesis, nucleoside and nucleotide degradation, polymeric compound degradation, and siderophore biosynthesis (S9_Table5 in S1 File).

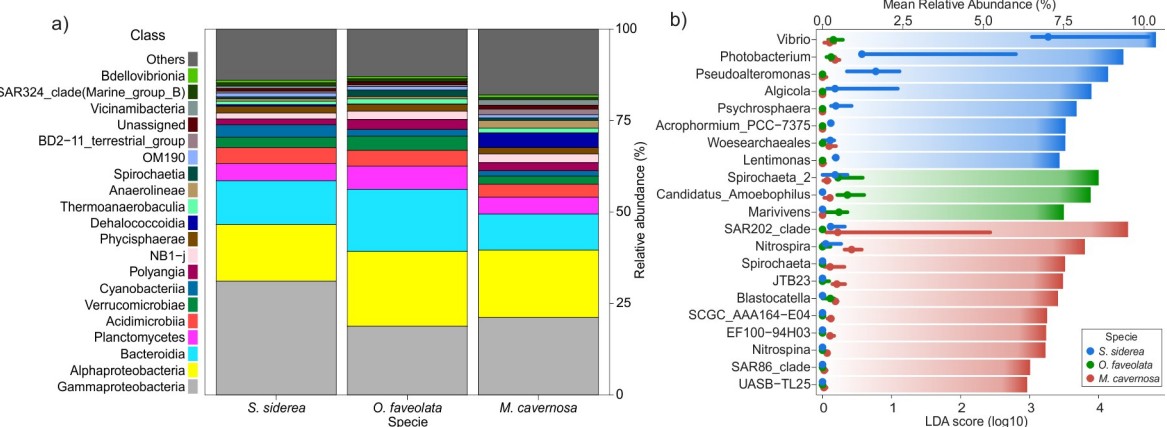

**Fig 2.** a) Microbial community composition of the three coral species healthy tissues from the Mexican Caribbean. b) LEfSe plot of the genera with differential abundances according to LDA score (bars), and their relative abundances (lines, median).

## Orbicella faveolata

Significant differences in microbial community structure were found in *O. faveolata* (PERMA-NOVA: F = 3.35, p = 0.002, Fig 3C), but not in alpha diversity (Shannon index: F = 3.57 p = 0.07, Fig 3A; richness: F = 2.97 p = 0.07, S1_Table1 in S1 File). The genera with differential abundances that had an effect on these changes in healthy tissues were *NB1-j*, *Roseibacillus*, *SAR324*_clade *(Marine_group_B)*, and *Dadabacteriales*, whereas in SCTLD tissues they were *Algicola*, *Pseudoalteromonas*, and *Marinifilum* (Fig 4B). Differences among metabolic potentials were not found (PERMANOVA: F = 0.8146, p = 0.489). The pathways related to chlorophyllide α biosynthesis, nitrogen, and sulfur metabolism were enriched in healthy tissues, whereas the best-represented pathways in SCTLD tissues were vitamin biosynthesis and sugar degradation. The list of pathway names is given in S10_Table6 in S1 File.

## Montastraea cavernosa

No significant differences between healthy and SCTLD tissues were found in alpha diversity (Shannon index: F = 4.36, p = 0.067, Fig 3A; richness: F = 4.20, p = 0.5, S1_Table1 in S1 File), microbial community structure (PERMANOVA: F = 1.0828, p = 0.355, Fig 3D), or the functional prediction (PERMANOVA: F = 0.6357, p = 0.568) in *M. cavernosa*.

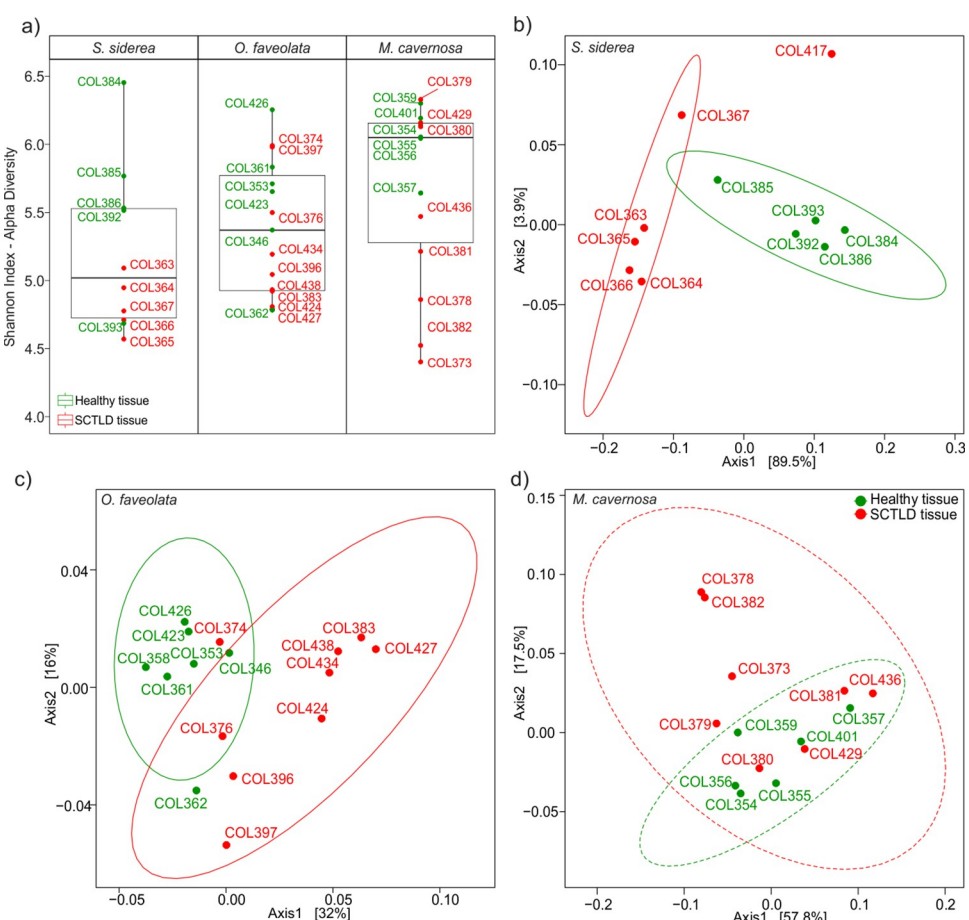

**Fig 3. Comparison between healthy and SCTLD tissues in three coral species.** a) Shannon index diversity, and DPCoA plots of b) *Siderastrea siderea*, c) *Orbicella faveolata*, and d) *Montastraea cavernosa*. Solid-line ellipses indicate a significant distribution of the variation between healthy and SCTLD tissues. Dashed-line ellipses are not significant in d.

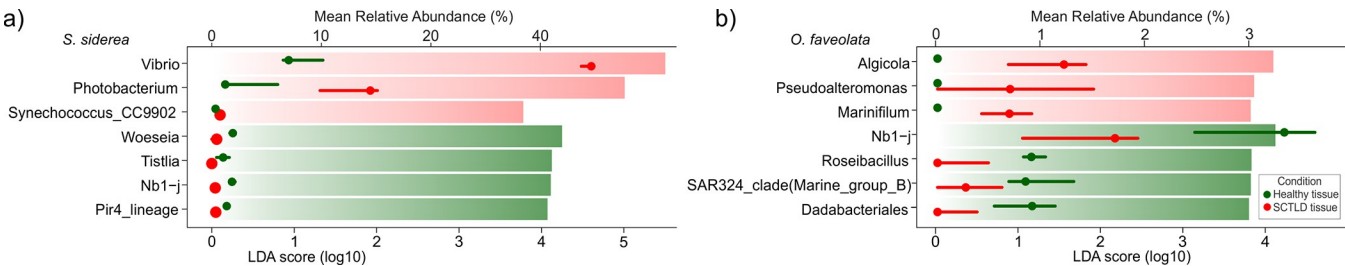

**Fig 4.** LEfSe plots showing the fourth genera with the highest LDA scores and mean relative abundances from a) *Siderastrea siderea* and b) *Orbicella faveolata*.

## Discussion

This study characterizes the microbiomes of three coral species and compares apparently healthy colonies of each species with SCTLD-affected colonies from the Mexican Caribbean. Our results showed that, while healthy corals exhibited similar diversity, composition, and predicted metabolic pathways, their bacterial community structures differed significantly. We also showed that the microbiome response to the presence of SCTLD varied by coral species: *S. siderea* displayed changes in diversity, structure, and function, whereas *O. faveolata* only showed structural shifts. Notably, *M. cavernosa* exhibited no microbiome differences between healthy and diseased tissues. These findings suggest that both coral species and health status significantly impact the microbiome's structure, diversity, and functional profiles during an SCTLD outbreak.

### Differences in microbial community structure from healthy coral tissues

In this study, the microbial community diversity (Shannon indices) did not show significant variation between all healthy samples of the three coral species (H' values from 4 to 6). This pattern has also been reported for the Florida region [15, 37]. Thus, a potential uniformity in alpha diversity within the studied coral communities is evidenced despite the potential limitations associated with sampling, sequencing, and bioinformatics protocols that can affect this diversity metric. *Gammaproteobacteria* and *Alphaproteobacteria* were the most abundant classes (~40%, Fig 2A) for healthy samples from all three coral species (Fig 2). Similar high abundances of these two classes have also been reported for *S. siderea* and *O. faveolata* in previous studies from Florida [37, 38], Puerto Rico [39], and Colombia [40]. Thereby, it suggests that the same coral species could share dominant microbial classes, as an important component of their core microbiome, despite the wide geographical separation of approximately 1,600km.

Specific differences in the community composition were also observed despite the presence of dominant groups in the microbiome of the same coral species from different regions. For example, *S. siderea* healthy tissues in this study exhibited *Desulfobactereota* and *Fusobacteriota* among the ten most abundant phyla, while Bonthond [37] did not, but reported *Nitrospirae* within their top ten, representing 3% of the microbiome. In contrast, in our samples, it only represented 0.3%. For *O. faveolata*, phylum *Mixococcota* and class NB1-j were identified as part of the most abundant groups (~3% each one) in this study, in contrast to Kimes et al., (2013), who did not find these groups in the same species of coral. These differences may reflect both the inherent niche-specific microbiome ("resident") and the dynamic influence of the surrounding environment ("variable") on coral holobionts, as has been reported in previous studies [3, 41–44].

Healthy tissues from all three coral species exhibited significant differences in community structure (PERMANOVA, p<0.01). Based on the community composition analyses, these

community structure differences could be related to the rare biosphere, defined as the large number of populations with very low abundances present in microbial communities [45]. The rare biosphere is highly divergent, holding a hidden reservoir of immense significance due to its high genomic diversity, serving as a source of innovation and resilience. Moreover, it can swiftly respond to changes in the host's physiology and environment, playing a crucial role in coral survival [41, 45, 46]. Notably, most members of the rare biosphere remain unidentified in this study, 5,533 out of 9,520 identified ASVs remained classified as "Unknown", which likely stems from the challenges associated with the culture of these microorganisms, their correct taxonomic assignment, and their ecological roles due to their low abundances.

Metabolic prediction analyses in this study indicated a common predicted pathway across all coral species involved in amino acid, carbohydrate, cell structure, and inorganic nutrient metabolism (S8_Table4 in S1 File). However, each species displayed unique metabolic enrichments and predicted metabolic functions, aligned with the abundance of specific bacterial groups. For example, in *M. cavernosa*, the most enriched genus was SAR202_clade (Fig 2B), which is known to participate in organic matter oxidation [47]. Additionally, metabolic pathways related to $CO_2$ fixation were better represented (S8_Table4 in S1 File), which aligns with Fiore [48], who reported a high abundance of carbon fixation genes in this coral species. This is in concordance between functions and groups of bacteria which have also been reported in other studies [22], which identified putative metabolic functions by comparing ASVs founded in *O. faveolata* tissues from the Mexican Caribbean, with the database Functional Annotation of Prokaryotic Taxa (FAPROTAX v.1.2.4) and with the current literature. Therefore, the relationship between the enriched bacterial groups and the enriched pathways in each coral species is expected, with the possibility of determining the specific microbial groups in each relationship by applying bioinformatic methods designed to make these determinations, since this relationship is not linear, because the same genes can be present in different bacterial groups.

## Specific response of microbial communities to SCTLD

Substantial differences in the microbial community between healthy and SCTLD tissues were found in this study and were consistent with previous studies [15, 16, 20]. The microbiome variations can result from stochastic events, leading to each modified microbiome being potentially unique [15]. We found that the microbial community followed some patterns according to the coral species which it is associated with, and the changes can be observed in different ways, such as in alpha diversity, composition, structure, or function of the microbiome, denoting a microbiome-wide shift that can be considered dysbiosis, or the disruption of the healthy microbiome. Whether this behavior is a cause or a consequence of SCTLD presence, is something we can not yet determine, and we believe that it is possibly negative feedback that undermines the vitality of the coral.

The increase of the microbial diversity associated with diseased communities has been observed and suggested to be associated with the increase of nutrient availability due to decomposing tissue which allows the faster growth of a wide range of opportunistic microorganisms [15, 39, 49]. Our findings showed no significant increase, but rather a potential decrease, in bacterial diversity for SCTLD-infected tissues. This reduction might be attributed to the detachment of infected tissue, leaving the exposed skeleton vulnerable to rapid algal colonization [5, 13, 50].

Alpha diversity among healthy tissues and SCTLD tissues was significantly different only in the microbiome associated with *S. siderea* (Fig 3A). This may be due to the drastic increase in the *Vibrionaceae* family. *S. siderea* had 279 families in their healthy tissue microbiome, and, in

the SCTLD samples, 182 of them disappeared while 15 appeared, highlighting the significant increment of the *Vibrionaceae* abundance, from a mean of 13.42% to 55.47%, with two genera being the most enriched, *Photobacterium* and *Vibrio*, the latter having increases of approximately 40% in the relative abundance of the samples.

The genus *Vibrio* is a cosmopolite genus present in riverine, estuarine, coastal, and marine ecosystems. They are chemoheterotrophic halotolerant microorganisms that grow in a 30–40°C optimal range, with a pronounced multiplication when the temperature increases up to 30°C [51]. As a matter of fact, SCTLD was first reported after a bleaching event related to an increase in sea surface temperatures in Florida [6, 7]. Moreover, *Vibrio* has been associated with a wide range of diseases of marine organisms, such as cnidarians, mollusks, crustaceans, and fish. In corals, it has been characterized as a related pathogen of the white plague [52], the black band [53], and rapid tissue loss [54].

Although our results agree with recent studies that also report *Vibrionaceae* with differential abundances, enriched in SCTLD samples [14, 15, 18, 20], the patterns are inconsistent in all tissue samples and coral species. Hence, they are unlikely primary pathogens and are more likely to be SCTLD opportunistic microbes. Although more in-depth studies are needed, focusing specifically on the *Vibrionaceae* family, is necessary to corroborate its role in the development of this disease. Given that *Vibrio* was also the most abundant genus in *S. siderea* healthy tissues (S11_Fig5 in S1 File), it is possible that the microbiomes of our samples were already different from that of corals where SCTLD had not yet arrived. Thus, our apparently healthy samples could represent an early stage of surface tissue loss, supporting the idea that corals may suffer a disturbance of their microbiomes prior to visible lesion formation [20].

Our results showed significant changes in the structure of bacterial communities between healthy and SCTLD tissues in both *S. siderea* and *O. faveolata*. This could be related to different events happening simultaneously, where the coral species could be displaying different responses according to their susceptibility to SCTLD [14, 15, 18, 20]. The healthy microbiome can be altered due to physiological changes in the diseased host and by direct effects of the pathogen, disrupting coral-bacterial interactions and changing the healthy microbiome structure [55, 56]. These structural changes could alter the physiological function of both the coral and the microbiome, potentially constraining beneficial processes, such as antibiotic production, weakening the natural protection of coral, and allowing opportunistic pathogens to colonize and further damage the coral. Furthermore, the disease creates a metabolic burden on both the coral and its microbiome, as metabolic resources of the entire coral colony will be allocated to confront the lesions caused by the disease.

This succession of events is regulated by the characteristics of the environment to which the coral colonies are exposed, their microbiome response, and the species' susceptibility level. For instance, *S. siderea* is a coral species that develops directly on the sediment, while *M. cavernosa* is found on the walls of the reef, and *O. faveolata* generally grows on the crest of the reef [12]. Therefore, exposure to currents, predators, and even sunlight differs slightly between the three species.

The cause of the different susceptibilities of each species is still unknown [14, 15]. However, it could be the coral offering resistance, its microbiome, or a mixture of both as an integral function of the holobiont. For example, the genus NB1-j appears as one of the genera with differential abundances with the largest effect size in the structure change of the microbial communities of *S. siderea* and *O. faveolata*, where it is enriched in healthy tissues of both species. In previous studies, it has been related to the degradation of hydrocarbons and the nitrogen cycle, postulating a possible association with *Ostreobium*, an alga associated with the coral skeleton, by providing nitrogen to the algae and obtaining organic carbon from them [57]. Another example is the genus *Nitrospira*, which is well represented in the samples of *M.*

*cavernosa*, the least susceptible of the coral species in this study [14, 15]. In *S. siderea*, *Nitrospira* is found in lower abundance in the SCTLD tissues, and even in the healthy samples, it did not have the abundance reported in the literature in the microbiome of this coral species, supporting the idea of an early stage of the disease in *S. siderea*. *Nitrospira* is a genus of nitrifying bacteria [58] that could also be related to maintaining the relationship between coral and its symbiotic algae, so its decrease could disrupt normal host-symbiont interactions, leading to the corals having diverse immune responses with not always favorable results [19].

Overall, this study aimed to identify microbes associated with three coral species and their changes due to the presence of SCTLD. We found that the microbiomes differed according to the species of coral with which they are associated, but characteristics such as their biodiversity and richness were similar, as well as part of their composition, sharing some bacterial groups. Just like corals, their associated microbiomes are affected at different levels by SCTLD, highlighting the potential link between microbiome structure and coral resilience against this disease. Future in-depth studies, encompassing field observations and controlled laboratory experiments, are crucial to unravel the specific mechanisms by which coral microbiomes influence resilience and potentially mitigate threats from diseases to reef continuity. This study contributes to the knowledge of the microbial ecology in coral reefs in the Mexican Caribbean and highlights the need to monitoring changes in bacterial communities when studying coral disease and stress events to achieve the maintenance, conservation, and restoration of coral reefs.

## Supporting information

**S1 File.**
(ZIP)

## Acknowledgments

The authors would like to thank Abril Gamboa-Muñoz, Esmeralda Perez-Cervantes, Francisco Puc-Itzá, and Jhonny G. García-Teh for lab and field assistance.

## Author Contributions

**Conceptualization:** Zita P. Arriaga-Piñón, Lorenzo Álvarez-Filip, Anastazia T. Banaszak, Ma. Leopoldina Aguirre-Macedo, David A. Paz-García, José Q. García-Maldonado.

**Data curation:** Zita P. Arriaga-Piñón.

**Formal analysis:** Zita P. Arriaga-Piñón.

**Funding acquisition:** Anastazia T. Banaszak, David A. Paz-García, José Q. García-Maldonado.

**Investigation:** Zita P. Arriaga-Piñón, Lorenzo Álvarez-Filip, Anastazia T. Banaszak, Ma. Leopoldina Aguirre-Macedo, David A. Paz-García, José Q. García-Maldonado.

**Methodology:** Zita P. Arriaga-Piñón, J. Eduardo Aguayo-Leyva.

**Project administration:** Anastazia T. Banaszak, David A. Paz-García, José Q. García-Maldonado.

**Resources:** Anastazia T. Banaszak, José Q. García-Maldonado.

**Software:** Zita P. Arriaga-Piñón, J. Eduardo Aguayo-Leyva, David A. Paz-García.

**Supervision:** Lorenzo Álvarez-Filip, Ma. Leopoldina Aguirre-Macedo, David A. Paz-García, José Q. García-Maldonado.

**Validation:** Zita P. Arriaga-Piñón, Lorenzo Álvarez-Filip.

**Visualization:** Zita P. Arriaga-Piñón.

**Writing – original draft:** Zita P. Arriaga-Piñón, José Q. García-Maldonado.

**Writing – review & editing:** J. Eduardo Aguayo-Leyva, Lorenzo Álvarez-Filip, Anastazia T. Banaszak, Ma. Leopoldina Aguirre-Macedo, David A. Paz-García, José Q. García-Maldonado.

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
