## [Decision Letter · Decision Letter 0]

11 Jul 2024

PONE-D-24-20565Microbiomes of three coral species in the Mexican Caribbean and their shifts associated with the Stony Coral Tissue Loss DiseasePLOS ONE

Dear Dr. García-Maldonado,

Thank you for submitting your manuscript to PLOS ONE. After careful consideration, we feel that it has merit but does not fully meet PLOS ONE’s publication criteria as it currently stands. Therefore, we invite you to submit a revised version of the manuscript that addresses the points raised during the review process.

We look forward to receiving your revised manuscript.

Kind regards,

Jiang-Shiou Hwang, Ph.D.

Academic Editor

PLOS ONE

Journal Requirements:

"This research was supported by Consejo Nacional de Ciencia y Tecnología (CONACYT) through grant FORDECYT-PRONACES, CF-2019-425888 to A.T.B., D.P.-G. and J.Q.G.-M. We thank CONAHCYT for providing the doctoral scholarship to ZPAP (830311) and JEAL (1081013) during the development of this study. "

"This research was supported by Consejo Nacional de Ciencia y Tecnología (CONACYT) through grant FORDECYT-PRONACES, CF-2019-425888 to A.T.B., D.P.-G. and J.Q.G.-M. We thank CONAHCYT for providing the doctoral scholarship to ZPAP (830311) and JEAL (1081013) during the development of this study. We also thank Abril Gamboa-Muñoz, Esmeralda Perez-Cervantes, Francisco Puc-Itzá, and Jhonny G. García-Teh for lab and field assistance."

"This research was supported by Consejo Nacional de Ciencia y Tecnología (CONACYT) through grant FORDECYT-PRONACES, CF-2019-425888 to A.T.B., D.P.-G. and J.Q.G.-M. We thank CONAHCYT for providing the doctoral scholarship to ZPAP (830311) and JEAL (1081013) during the development of this study. "

5. We note that [S2_Fig 1] in your submission contain [map/satellite] images which may be copyrighted. All PLOS content is published under the Creative Commons Attribution License (CC BY 4.0), which means that the manuscript, images, and Supporting Information files will be freely available online, and any third party is permitted to access, download, copy, distribute, and use these materials in any way, even commercially, with proper attribution. For these reasons, we cannot publish previously copyrighted maps or satellite images created using proprietary data, such as Google software (Google Maps, Street View, and Earth). For more information, see our copyright guidelines: http://journals.plos.org/plosone/s/licenses-and-copyright.

a. You may seek permission from the original copyright holder of S2_Fig 1 to publish the content specifically under the CC BY 4.0 license.  

Reviewers' comments:

Reviewer's Responses to Questions

**Comments to the Author**

1. Is the manuscript technically sound, and do the data support the conclusions?

Reviewer #1: Yes

Reviewer #2: Yes

2. Has the statistical analysis been performed appropriately and rigorously? 

Reviewer #1: Yes

Reviewer #2: I Don't Know

3. Have the authors made all data underlying the findings in their manuscript fully available?

Reviewer #1: Yes

Reviewer #2: Yes

4. Is the manuscript presented in an intelligible fashion and written in standard English?

Reviewer #1: Yes

Reviewer #2: Yes

5. Review Comments to the Author

Reviewer #1: Dear Authors,

The current study presents difference in diversity, composition and functional differences of microbial communities of 3 healthy and diseased coral species.

This paper is of sufficient scientific interest and originality in its contents. The authors have written it very well. Especially the discussion part is the strength of this manuscript.

Reviewer #2: It is not clear as to how amplicons from samples at various sites were pooled to produce sequencing libraries. How many samples contributed to each library?

What steps were performed to equalize sampling effort across libraries (rarefaction, etc.) Large read differences can result in incorrect diversity estimates and comparisons among samples. Please clarify.

Minor: Typo in line 323- "can be result" should be "can result"

6. PLOS authors have the option to publish the peer review history of their article (what does this mean?). If published, this will include your full peer review and any attached files.

Reviewer #1: **Yes: **Jojy John

Reviewer #2: **Yes: **William B. Schill

---

## [Author Response · Author response to Decision Letter 0]

31 Jul 2024

Comment No 1: Please ensure that your manuscript meets PLOS ONE's style requirements. 

A: We have made an additional revision of style requirements, and the manuscript meets the journal requirements. 

Comment No 2: In your Methods section, please provide additional information regarding the permits you obtained for the work. Please ensure you have included the full name of the authority that approved the field site access and, if no permits were required, a brief statement explaining why. 

A: Authority and permit were added.

“under collection permit issued to Universidad Nacional Autónoma de México (PPF/DGOPA-070/20).”

Comment No 3: Thank you for stating the following financial disclosure. Please state what role the funders took in the study. If the funders had no role, please state: "The funders had no role in study design, data collection and analysis, decision to publish, or preparation of the manuscript."

A: The funding institution, Consejo Nacional de Ciencia y Tecnología (CONACYT), had no role in study design, data collection and analysis, decision to publish, or preparation of the manuscript.

Comment No 4: Thank you for stating the following in the Acknowledgments Section of your manuscript. We note that you have provided funding information that is not currently declared in your Funding Statement. Please include your amended statements within your cover letter; we will change the online submission form on your behalf.

A: We eliminate the funding information from Acknowledgments Section. All the funding information was declared in the funding statement. 

Comment No 5: We note that [S2_Fig 1] in your submission contain [map/satellite] images which may be copyrighted. We require you to either (1) present written permission from the copyright holder to publish these figures specifically under the CC BY 4.0 license. 

A: We have provided information regarding the terms of use information for the map.

L616: “Data sources: (Administrative boundary layer: GADM database (www.gadm.org) under CC BY 4.0 license (https://gadm.org/license.html). The original maps were modified to show the study sites.”

Comment No 6: Please review your reference list to ensure that it is complete and correct.

A: A revision was made, and no retracted papers have been cited. The reference #18, in L480-482, was actualized.

Review Comments to the Author

Reviewer #1: The current study presents difference in diversity, composition and functional differences of microbial communities of 3 healthy and diseased coral species.

This paper is of sufficient scientific interest and originality in its contents. The authors have written it very well. Especially the discussion part is the strength of this manuscript.

A: Thank you for your comments. 

Reviewer #2: It is not clear as to how amplicons from samples at various sites were pooled to produce sequencing libraries. How many samples contributed to each library?

A: In this study each library was prepared by sample and different samples from various sites weren’t pool to produce sequencing libraries. To avoid confusion, this information was modified. L126 and L136: “Library preparation per sample …” and “Sequencing libraries for the 16S rRNA gene were prepared according to the manufacturer’s protocol.”

What steps were performed to equalize sampling effort across libraries (rarefaction, etc.) Large read differences can result in incorrect diversity estimates and comparisons among samples. Please clarify.

A: We recognize that large differences can affect the diversity metrics. Thus, in this study the data set was normalized, as indicate L151-152: “The data set was normalized with the phyloseq library to obtain abundance tables…”

Minor: Typo in line 323- "can be result" should be "can result"

A: Done, this was corrected in L322.

---

## [Editor Report · Decision Letter 1]

6 Aug 2024

Microbiomes of three coral species in the Mexican Caribbean and their shifts associated with the Stony Coral Tissue Loss Disease

PONE-D-24-20565R1

Dear Dr. García-Maldonado,

We’re pleased to inform you that your manuscript has been judged scientifically suitable for publication and will be formally accepted for publication once it meets all outstanding technical requirements.

Kind regards,

Jiang-Shiou Hwang, Ph.D.

Academic Editor

PLOS ONE
---

## [Editor Report · Acceptance letter]

15 Aug 2024

PONE-D-24-20565R1 

PLOS ONE

Dear Dr. García-Maldonado, 

I'm pleased to inform you that your manuscript has been deemed suitable for publication in PLOS ONE. Congratulations! Your manuscript is now being handed over to our production team.

Kind regards, 

on behalf of

Prof. Jiang-Shiou Hwang 

Academic Editor

PLOS ONE